# Acceptability of Clinical Trials on COVID-19 during Pregnancy among Pregnant Women and Healthcare Providers: A Qualitative Study

**DOI:** 10.3390/ijerph182010717

**Published:** 2021-10-13

**Authors:** Elena Marbán-Castro, Clara Pons-Duran, Laura García-Otero, Haily Chen, Luis Bernardo Herrera, María del Mar Gil, Anna Goncé, Elena Ferriols-Pérez, Miguel Ángel Rodríguez, Paloma Toro, Azucena Bardají, Raquel González, Clara Menéndez, Cristina Enguita-Fernàndez

**Affiliations:** 1ISGlobal, Hospital Clínic-Universitat de Barcelona, 08036 Barcelona, Spain; elena.marban@isglobal.org (E.M.-C.); clara.pons@isglobal.org (C.P.-D.); laura.garcia@isglobal.org (L.G.-O.); haily.chen@isglobal.org (H.C.); luisbernardo.herrera@isglobal.org (L.B.H.); azucena.bardaji@isglobal.org (A.B.); raquel.gonzalez@isglobal.org (R.G.); clara.menendez@isglobal.org (C.M.); 2Consorcio de Investigación Biomédica en Red de Epidemiología y Salud Pública (CIBERESP), 28029 Madrid, Spain; 3Obstetrics and Gynecology Department, Hospital Universitario de Torrejón, Torrejón de Ardoz, 28850 Madrid, Spain; mgil@torrejonsalud.com; 4School of Medicine, Universidad Francisco de Vitoria, 28223 Madrid, Spain; 5Fetal Medicine Research Center, BCNatal—Barcelona Center for Maternal-Fetal and Neonatal Medicine (Hospital Clínic and Hospital Sant Joan de Déu) and Institut d’Investigacions Biomèdiques August Pi i Sunyer (IDIBAPS), 08036 Barcelona, Spain; agonce@clinic.cat; 6Department Gynaecology and Obstetrics, Parc de Salut Mar University Hospital, 08003 Barcelona, Spain; eferriols@parcdesalutmar.cat; 7HM Puerta del Sur, Móstoles, 28938 Madrid, Spain; mrodzambrano@gmail.com; 8Obstetrics and Gynecology Department, Complejo Asistencial de Segovia, Hospital General de Segovia, 40002 Segovia, Spain; ptorop@saludcastillayleon.es; 9Centro de Investigação em Saúde de Manhiça (CISM), Maputo 1929, Mozambique

**Keywords:** COVID-19, pregnancy, acceptability, clinical trial, perceptions, healthcare professionals

## Abstract

Participation of pregnant women in clinical trials entails challenges mainly related to concerns about the risks for fetuses. We undertook a qualitative study from June to October 2020 to assess the acceptability of participating in COVID-19 clinical trials among pregnant women in Spain. Phenomenology and grounded theory were used as methodological approaches. Semi-structured interviews were conducted with 24 pregnant women and six healthcare providers. Women were unsure if pregnancy was a risk factor to acquire the infection or to develop severe disease and expressed the limited information they had received, which led to uncertainties and emotional suffering. They had concerns regarding participation in clinical trials on COVID-19, regardless of the drug under study. Healthcare providers alluded to the importance of involving pregnant women’s relatives at the recruitment visit of the clinical trial. These findings may be useful to facilitate pregnant women’s participation in clinical trials.

## 1. Introduction

More than half of pregnant women take drugs for the prevention or treatment of diseases, yet they are systematically excluded from clinical trials of medicinal products due to safety concerns, mostly related to potential harms to their fetuses [1,2,3]. Thus, it is rare that new investigational products are designed and evaluated for their use during pregnancy [1], and the majority of therapeutic products prescribed during pregnancy have not been adequately studied for this purpose [4]. In recent health crises, such as the Ebola epidemic, despite high maternal and perinatal mortality rates, pregnant women were excluded from drug and vaccine trials [5,6]. Similarly, this was the case during the Zika virus (ZIKV) epidemic despite the consequences of infection for maternal and child health [4,7,8,9]. Since results from studies conducted in non-pregnant populations cannot be directly extrapolated to pregnant women, researchers have advocated for the inclusion of pregnant and lactating women in drug and vaccine trials during the COVID-19 pandemic [1,2,10,11,12,13,14,15]. However, by July 2020, only 40 out of the 5492 registered studies on COVID-19 included pregnant women. From those 40 studies, only 5 of them were randomized controlled trials designed to evaluate treatments in pregnant women with COVID-19 [16].

Acceptance of participation in clinical trials and adherence to interventions is generally challenging. Enrolment of participants may be hindered by hesitancy on the safety of taking a new medicinal product, a lack of interest or distrust in the research, or discomfort from having extra appointments [17]. Clinical trials involving pregnant women face additional enrolment and engagement challenges, such as a greater reluctance in taking any drug during gestation and perceived potential negative consequences on fetuses [18,19]. Some of the factors influencing pregnant women’s decisions to participate in a clinical trial might be a perceived personal benefit, quality of the information received, including communication on the risks and potential complications, the role of relatives in the decision-making process, the perceived voluntariness, uncertainties due to randomization as part of the research design, and unwillingness to take the study medication [19,20,21,22]. A qualitative study about the acceptability of a hypothetical ZIKV vaccine showed that women who had had a previous ZIKV infection during pregnancy hesitated to participate in a clinical trial; however, they would accept vaccination once the vaccine was commercialized [23].

The limited number of clinical trials on investigational medicinal products for COVID-19 treatment and prevention during pregnancy highlights the need to engage pregnant women in research to develop evidence-based recommendations [11]. Yet, efforts to understand pregnant women’s experiences regarding their participation in clinical trials should also be encouraged [21]. Furthermore, it is known that enrolment of pregnant women in clinical research can be influenced by perceptions and experiences of clinicians and healthcare providers delivering maternal healthcare; thus, examining their views on this issue should also be considered [24,25].

The objective of this study was to assess the acceptability of participating in a hypothetical clinical trial on COVID-19 during pregnancy. We explored perceptions on the risks for SARS-CoV-2 infection and severity of COVID-19 disease, as well as barriers and facilitators to participate in a COVID-19 clinical trial among pregnant women and healthcare providers to help improve strategies for engagement of pregnant women in clinical trials. The study may contribute to the understanding of the impact of COVID-19 on women pregnant during the pandemic.

## 2. Materials and Methods

### 2.1. Study Design and Population

This was a qualitative study based on phenomenology (methodological approach to understand first-hand experiences) [26], and grounded theory (inductive approach to analyze data, where theoretical generalizations emerge from it) [27]. The study design was informed by a definition of acceptability referred to as “a multifaceted construct that reflects the extent to which people delivering or receiving a healthcare intervention consider it to be appropriate, based on anticipated or experienced cognitive and emotional responses to the intervention” [28]. The study was conducted in Spain, between July and October 2020, within the frame of a clinical trial assessing the efficacy of hydroxychloroquine (HCQ) in preventing SARS-Cov-2 infection and severity of COVID-19 during pregnancy and postpartum [29]. The study was performed at the end of the first COVID-19 pandemic wave and the beginning of the second wave in the country.

Study participants were pregnant women and healthcare providers who attended pregnant women. Inclusion criteria for pregnant women were defined as being pregnant and willing to be interviewed, regardless of their eligibility to participate in the above-mentioned clinical trial. Invited healthcare providers included both medical doctors and nurses working at the Obstetrics and Gynaecology Departments of study hospitals. Study participants were mainly identified from four out of the nine hospitals participating in the clinical trial in Barcelona (Hospital Clínic de Barcelona, and Hospital del Mar) and Madrid (Hospital Universitario de Torrejón de Ardoz, and HM Puerta del Sur, Móstoles); participation in the clinical trial was not a requirement for inclusion in the interviews. Initial sampling was purposive, and snowball sampling was also applied to interview other pregnant women who were not necessarily attending study hospitals. In addition, convenience sampling was applied to facilitate access to women willing to be interviewed. The sample size was defined based on a saturation point whereby all themes had been thoroughly explored and no new themes were emerging in subsequent interviews [26,27,30].

### 2.2. Data Collection

Data were collected through semi-structured interviews and field notes from interviewers [27]. The design of the questions’ guide was based on three topics of interest, namely, (1) *knowledge about SARS-CoV-2 infection and COVID-19*, (2) *lived experience of the pandemic during pregnancy*, and (3) *women’s understanding and hypothetical participation in clinical trials*. The topic guide was pilot-tested to check the interviewing style and approach (see Appendix A). Interviews were performed in Spanish or Catalan and lasted approximately 20–40 min each. Before the interview took place, sociodemographic information from participants was collected and registered in an Excel sheet. Two researchers performed the interviews (E.M.-C. and C.E.-F.). Each interview was performed by one of the researchers and coded by the other researcher to gather more detailed and in-depth information. All interviews were carried out remotely by video call through online platforms or phone calls, according to participants’ preference, and at the time chosen by them. All interviews were audio-recorded, and notes were taken. Researchers regularly discussed key findings, difficulties, and any changes to the data-collection guides according to emerging data. All participants seemed comfortable during the interviews, willing to share their experiences, and participate in-depth in the discussions.

### 2.3. Data Analysis

Audio recordings were summarized, and particular pieces related to the topics of interest were transcribed verbatim by the interviewers. Thematic analysis was performed, and data were manually coded. An initial coding frame was developed based on themes arising from the analyses of the first transcripts using open and inductive coding. Then, consensus on codes and emerging themes were reached between two investigators (E.M.-C. and C.E.-F.). A final coding frame was agreed upon, and themes and categories were grouped with verbatim quotes. This article has been prepared as per reporting standards set forth in the SRQR guidelines for reporting qualitative studies [31] (see Appendix A).

### 2.4. Ethical Considerations

Ethical approval for the study was granted by the Ethics Review Committee of the HCB, Barcelona, Spain (Reg. No. HCB/2020/0382). The study was conducted in accordance with the Good Clinical Practice Guidelines and under the provisions of the Declaration of Helsinki and local rules and regulations. Participants gave oral consent for interviews and audio recording [32]. Their oral permissions were also audio-recorded before the interview began. All names and other personal identifiers in the transcripts were deleted to guarantee subject anonymity.

## 3. Results

Women’s acceptability to participate in a COVID-19 clinical trial during pregnancy included different themes that emerged from the interviews conducted with pregnant women and healthcare workers, such as: *knowledge and uncertainties about COVID-19 and pregnancy, perceptions of disease severity and vulnerability, pregnant women’s emotional state during the pandemic*, and *barriers and facilitators to trial participation*.

### 3.1. Participants Characteristics

A total of 30 participants were included in the interviews, 24 pregnant women and 6 healthcare workers. Six women self-reported to have had COVID-19 during pregnancy, and one woman before pregnancy. Only one interviewee was participating in the clinical trial, and another one declined the invitation to participate. Sociodemographic characteristics of participants are presented in Table 1.

### 3.2. Knowledge and Uncertainties about COVID-19 and Pregnancy

Concerning COVID-19 symptoms, modes of transmission, and potential treatments, pregnant women’s general knowledge was similar to the information from the mainstream media. The majority of respondents mentioned the most common symptoms associated with COVID-19 disease (cough, fever, respiratory distress, fatigue, and loss of smell and taste). Women described symptoms of COVID-19 that were similar to those experienced by the general population, normally referred to as a flu-like disease. Knowledge of COVID-19 treatment options was limited.

The lack of an effective treatment for COVID-19 was recognized by the interviewed healthcare professionals. Pregnant women declared not to have heard of any effective drug to treat the infection. Some women mentioned to have heard about COVID-19-related drugs on the news. Interviewed women mentioned HCQ to be ”the drug that Trump takes”, the one causing “stock-outs in the USA”, or ”the drug not recommended to be used by the WHO”, which led to negative perceptions of HCQ. Other participants mentioned that researchers are also using ‘the drug for HIV/AIDS’ to treat COVID-19.


*“The first thing I found when I searched for it [HCQ in Google] was ‘The WHO discourages [its use]’ (…). They are proposing to a pregnant woman one thing [participation in a clinical trial] that WHO is saying ‘no’. I was a bit indignant”*
(Pregnant woman, 43 years old).

Generally, women complained not having received detailed information about COVID-19 during pregnancy. Women doubted whether or not they were a vulnerable group for COVID-19 and whether or not they needed to take additional measures to prevent the infection.


*“I did not find any information [about COVID-19 and pregnancy] that seemed to be accurate or that could contribute with more information than this [information found in the media]”*
(Pregnant woman, 27 years old).

The most common sources of information referred by participants were: (1) TV news; (2) social media; (3) search results on the internet; (4) counsel from healthcare professionals; (5) opinions from friends or relatives that worked in medical or scientific fields; and (6) information from other people, such as neighbors.

Women repeatedly stated to have received little and poor-quality information on COVID-19 in pregnancy during their routine maternity visits and that the limited frequency of in-person medical visits could have worsened this communication channel. The majority of women were unaware of the possibility of vertical transmission of SARS-CoV-2 virus; some women mentioned that they were aware of some cases of health problems in pregnancy that appeared on the news, but they were not sure if they were real. Some women affirmed that their concern on the possibility of vertical transmission and clinical manifestations in fetuses increased with increasing scientific evidence.


*“I know that…normally, it’s a risk group [pregnant women], but I don’t know to what extent it [COVID-19] could infect the fetus”*
(Pregnant woman, 41 years old).

Uncertainties regarding COVID-19 information resulted in two different behavioral pathways: (1) some women looked for information on the disease and possible consequences for fetuses either on the internet and/or asking healthcare staff, as they believed that “to have information, reassures me”; or (2) stopped looking for information, since it was alarming and they were afraid to know the consequences (avoid reading news about pregnancy and/or COVID-19, etc.).


*“I didn’t google for ‘pregnancy and COVID what can happen’, because then maybe I would be scared…of not even wanting to go out from home”*
(Pregnant woman, 35 years old).

### 3.3. Perceptions of Disease Severity and Vulnerability

The limited information received on COVID-19 in pregnancy led to misconceptions in the way women perceived risks regarding the infection. Some women were convinced that given the unavailability of treatments, they could suffer severe COVID-19. However, other women mentioned that the risk of contracting the infection and/or developing severe symptoms was lower in pregnancy due to the fact that pregnant women are generally young, have more regular medical check-ups, generally take more preventive measures, and have a stronger immune system.


*“I read that it [COVID-19] affects pregnant women the same as other people… there’s not a higher risk [among pregnant women]”*
(Pregnant woman, 43 years old).

Two women mentioned that the main risk of COVID-19 during pregnancy was not the infection itself but the lack of treatments for them once infected. Despite this, this reasoning was not identified among their arguments for considering their potential participation in a COVID-19 clinical trial.

Some women disliked being labeled as “vulnerable” for being pregnant. Additionally, some participants mentioned that despite being considered as a “vulnerable group”, no clear actions were taken accordingly. Thus, this “label” was perceived as not translated into specific recommendations to prevent the infection and deconfinement-specific measures, as was the case for children and the elderly during the de-escalation plan established after lockdown by the Ministry of Health in Spain.

### 3.4. Pregnant Women’s Emotional State during the COVID-19 Pandemic

Pregnant women’s emotional state during the pandemic emerged as a relevant theme to further understand the decision-making process concerning participation in COVID-19 trials during pregnancy. Most women affirmed that they had experienced fear for the unknown, for upcoming COVID-19 outbreaks in case they still are pregnant, and on the effects on their children. They mentioned they have felt lonely since medical protocols obliged them to attend routine antenatal care visits without their partners. Two out of the six women diagnosed with COVID-19 mentioned having felt left apart and even suffered social stigma. Women’s accounts were also mirrored in healthcare providers’ insights.


*“[It was] a health emergency, you saw that the information was changing (…) because of course, it was a new virus, things were changing, so what a lack of coordination! (…). We were facing a pandemic, the unknown, something never experienced, at least for me, a nurse that, I have to say, I have not been here (working) for two days (…) and I have never experienced this, never”*
(Healthcare provider).

In general, healthcare providers mentioned that they also had little information about COVID-19 in pregnancy, and how this led to concerns about the provision of adequate care for their patients; they also feared for their health. Their main concern was the limited information they had to respond to pregnant women’s doubts and questions.

Some pregnant women expressed that the lockdown period had affected their mental health. In general, women felt safer while being at home because restrictions made it difficult to be infected. However, during the de-confinement, they had to make decisions about seeing friends and relatives. Pregnant women carried the burden of decision making and became anxious when thinking they could become infected.


*“I’m having a worse time now, with this ‘new normal’, than during the lockdown (…) and now I feel distressed because I don’t see the end. So, I see that people is losing fear…”*
(Pregnant woman, 43 years old).

### 3.5. Barriers and Facilitators to Participating in a Clinical Trial during Pregnancy

Only a third of the women interviewed said that they would participate in a clinical trial when non-pregnant, but this willingness to participate was even lower during pregnancy (16.7%); during lactation, it was even lower. These proportions were also observed in female healthcare providers’ responses regarding their own participation in a clinical trial during pregnancy. However, their acceptability to participate while lactating was much higher.

Pregnant women’s responses about their participation in a clinical trial on HCQ did not differ if the clinical trial was on a different drug (such as remdesivir, dexamethasone, or others). The study drug seemed not to be a factor to determine pregnant women’s acceptability to participate in the trial, but it was a facilitator to participate for healthcare providers. Table 2 illustrates responses given by study participants regarding acceptability in clinical trials.

Pregnant women accounted for several reasons for their acceptance or refusal to participate in a hypothetical clinical trial on COVID-19; in case of refusal, they also listed the possible circumstances that could modify their choice. Reasons given by participants are illustrated in Table 3.

According to pregnant women’s reasoning for accepting, or not, to participate in a clinical trial, several barriers and facilitators were identified and can be classified into individual, interpersonal, and trial-related factors, as shown in Table 4.

Rejection to drug uptake during pregnancy and fears of adverse effects on the babies were pointed as the major reasons for refusing to participate in a clinical trial. Healthcare professionals explained that women often referred to “secondary effects related to the drug” as a barrier to participate in the trial. However, when exploring the reasons for not participating in the trial in depth, women mentioned “adverse effects related to potential fetal damage”, using both terms interchangeably.

Healthcare providers’ concerns for inclusion of pregnant women in a clinical trial are likely to negatively affect recruitment. During the interviews, healthcare providers expressed doubts regarding their own potential involvement in a clinical trial as study subjects. One health professional acknowledged that these doubts could influence the recommendations given to pregnant women to participate in a trial. Some healthcare providers mentioned that they would not recommend their patients to be enrolled in a clinical trial if they themselves do not trust the drug, the clinical trial design, or the research team.


*“To recommend something with which I do not feel good…I see it…a bit…incorrect. If it is a drug that does not have risks, I would take it! But me, in the moment that it has some risks, I would recommend it to my patients, but I would not hide those risks”*
(Healthcare professional).

Regarding facilitators for clinical trial participation, pregnant women expressed that they would change their decision if a direct benefit to themselves or their babies could be perceived, in the case of COVID-19, this would be a reduced risk of infection and/or vertical transmission.


*“If I’m being told that the virus can brutally affect the fetus…this is another thing, huh? (…) Maybe I’d tend a little more towards the ‘yes’, huh?”*
(Pregnant woman, 35 years old).

Healthcare providers mentioned the presence of pregnant women’s partners’ or relatives’ in the recruitment process as facilitators for trial participation acceptability. This was also reported by some of the pregnant women interviewed. Both healthcare providers and the pregnant women themselves acknowledged that recommendations given to pregnant women regarding generally avoiding taking drugs hinder their participation in clinical trials on medicinal products.


*“I can’t take medicines [because of being pregnant]), and now it turns out that mmm… they have to give me drugs for something else…? (…) No, no…”*
(Pregnant woman, 35 years old).

Healthcare providers also mentioned that the limited available information on COVID-19 in pregnancy might also affect pregnant women’s decision to participate in a clinical trial on the infection. Healthcare workers also differentiated the willingness to participate in the clinical trial between participants who have COVID-19 and those who are uninfected or asymptomatic.

Based on the identified barriers and facilitators, different factors emerged as affecting the decision-making process of pregnant women to participate in a clinical trial, as illustrated in Figure 1.

## 4. Discussion

In this study, we have identified some potential barriers and facilitators as well as explored pregnant women’s and healthcare providers’ perceptions and experiences of participating in a drug-based clinical trial during pregnancy during the COVID-19 pandemic in Spain. Our findings point to several factors that influence the decision-making process to participate in clinical trials during pregnancy, which, in turn, dialog with the multidimensional approach to acceptability that has informed our study [28]. The theoretical framework proposed by Sekhon et al. [28] envisions seven component constructs comprising a complex conceptualization of acceptability (either anticipated or reported). Even if in our study factors have been grouped in broader categories, certain parallels can be found between the identified potential barriers and facilitators and the seven constructs that build up the definition of acceptability in the aforementioned theoretical framework. For instance, the affective attitude and the burden associated with participating in the intervention are reflected in our participants’ insights. Therefore, our results support the need to adopt a multidimensional perspective to provide a more nuanced assessment of pregnant women’s acceptability to participate in clinical trials.

In general, pregnant women’s acceptability to participate in a clinical trial on COVID-19 was low, and concerns and fears around potential harm to their babies were the main reasons for it. Most women would not accept participating in a clinical trial, but they would accept it in the case where a personal benefit was perceived. For instance, women reported they would participate in a trial if vertical transmission of the virus is evidenced to be common or if the disease in pregnancy entails severe risks and complications. Thus, the individual risk–benefit assessment seems to be decisive in the context of our study. As already reported, helping others or advancing science are not strong enough arguments for improving pregnant women’s acceptability [24].

Factors influencing pregnant women’s decisions regarding clinical trial participation have been previously identified [18,19,20,21,22]. However, the context of the COVID-19 pandemic could have accentuated the effect of these factors on pregnant women’s decision-making processes. It has been argued that contexts and circumstances in which individuals are asked to consider their participation in a clinical trial are determinants influencing their decision-making [25]. In the context of the COVID-19 pandemic, factors related to uncertainty, a perceived lack of access to entitled sources of information, and mental stress and fear could all be underpinning the decision to participate in a clinical trial during pregnancy. We have observed in this study that the context of the COVID-19 pandemic has had a profound effect on the emotional state of pregnant women, which has framed their overall reaction to participating in a COVID-19 clinical trial, thus, influencing their anticipated acceptability to participate [28]. Interviewed women lacked quality information about COVID-19 in pregnancy from the media, and importantly, from their healthcare providers. The latter influenced the experience of anxiety and emotional distress that increased after the ending of the lockdown period due to the perception of increased risks of being infected during the de-escalation of control measures. However, the changes in women’s risk perceptions did not lead to changes in their attitudes towards potential participation in a clinical trial. The emotional burden of being pregnant during a pandemic plus the general sense of uncertainty about the disease is likely to influence women’s willingness to participate in a clinical trial. Similarly, our findings also suggest that close relatives and partners have an important role to play in the decision-making process, in contradiction with a study conducted in the UK pre-COVID-19 pandemic, where the influence of families on pregnant women’s decisions on trial participation was apparently minor [21].

We also observed that the study drug did not seem to influence the acceptability of trial participation. These findings are in agreement with other studies and suggest that the main concern about participation in a clinical trial relates to the potential consequences of the drug on the fetus, regardless of the type of drug [33]. Nevertheless, the contrast between the attitudes of healthcare professionals towards the use of certain drugs during pregnancy and the initial reluctance to take drugs in the context of a clinical trial, as expressed by pregnant women, suggests that the fear of taking drugs during pregnancy probably relates to the insufficient information on the therapeutic value of certain drugs in pregnancy. Pregnant women were more prone to participate in a clinical trial when it was clearly explained that the drug under study was already used during pregnancy [34].

A cross-cutting response, confirmed by the views of the healthcare providers interviewed, was the need for the women to be reassured that there would not be any potential harm to the fetus when invited to participate in a clinical trial. This leads to reflection on how potential risks of participation are transmitted during recruitment processes in clinical trials in pregnancy. On the one hand, potential adverse effects to the fetus (preterm birth, spontaneous abortion, fetal or child anomalies, etc.) are not the same as secondary effects that the woman may experience (dizziness, vomiting, headache, etc.). The interchangeable use of these terms leads to confusion between *secondary effects*, inherent in all drugs, and *adverse effects*, which might increase hesitancy to participate in the clinical trial. On the other hand, the potential risks of participation in clinical trials as determined by researchers may not match pregnant women’s understandings and perceptions of these risks, which is a challenge to deal with during the recruitment processes [24]. Ethical concerns may arise when defining an “acceptable risk” a woman must accept during their participation. It may differ due to different women’s “reasonable risk perception” and the extent to which these risks could be tolerated [4]. All these concerns must be considered when tailoring the messages at the time of recruitment.

The findings from this study also suggest that clear messages might not be enough to facilitate pregnant women’s participation. In alignment with other studies, most of the interviewed women expressed the need to count on known and trusted sources of information, such as obstetricians and midwives [21]. Women’s responses can be explained by the fact that they were not part of the clinical trial within which the study was framed. Their answers referred to hypothetical participation, hence not being aware that enrolled participants actually counted on these sources at the time of recruitment. Nonetheless, their perspectives point to the right of pregnant women to decide about their participation in a trial after proper counseling from healthcare staff so that all possible risks and benefits of their participation are mentioned [18,19,22,23]. Therefore, there is a need to ensure an accurate communication channel in an empathetic atmosphere during the recruitment processes, especially in the delicate context of the COVID-19 pandemic.

Despite the need to include pregnant and lactating women in clinical research [11], approaches to enhance adherence to their participation seem to still be overlooked. As illustrated in Figure 1, deciding to participate in a clinical trial during pregnancy can be influenced by a combination of factors, the relevance of which may vary depending on the circumstances, from emergencies to situations where decisions can be taken calmly. Thus, a contextually sensitive understanding of pregnant women’s perceptions, attitudes, and behaviors is key for the correct and ethical success of the clinical development of medicinal products, including clinical trials and their upcoming implementation.

The main limitation of the study is that a desirability bias might have affected the results, as healthcare professionals responsible for trial recruitment could be less likely to respond honestly to certain questions since they participated in a trial on HCQ in pregnancy. The main strengths of the study are the accounting for pregnant women’s experiences, which may contribute to improving research procedures, as well as the feedback and insights provided directly by affected populations, women who faced the challenges of the COVID-19 pandemic while pregnant, and their healthcare professionals. Nevertheless, healthcare providers’ perceptions and first-hand experiences on maternal care during the pandemic provided valuable insights. By analyzing these dual perspectives, we think this study has contributed to a more nuanced understanding of the factors that need to be considered during trial’s recruitment processes involving pregnant women, especially in a context such as the COVID-19 pandemic. Finally, by providing relevant insights on how pregnant women relate to clinical research, the study also points at crucial issues to analyze conceptualizations of motherhood and pregnancy or ethics in research during pregnancy.

## 5. Conclusions

Acceptability to participate in a drug-based clinical trial on COVID-19 during pregnancy among pregnant women in Spain during the COVID-19 pandemic context was generally low. The reasons for this may be due to the limited knowledge about COVID-19 and its consequences in pregnancy, together with a low perception of vulnerability. The most important facilitators found were trust in healthcare providers and the understanding that the study drug would not cause harm to the fetus. The role of partners and relatives may be relevant in the decision-making process. Clinical trial participation might be boosted by facilitating an effective communication strategy among healthcare professionals and women, responding to their doubts regarding drug safety and its effects on fetal health, and including other family members in this process. Further operational research is needed to understand the perceptions of pregnant women about medical products that may be needed during pregnancy.

## Figures and Tables

**Figure 1 ijerph-18-10717-f001:**
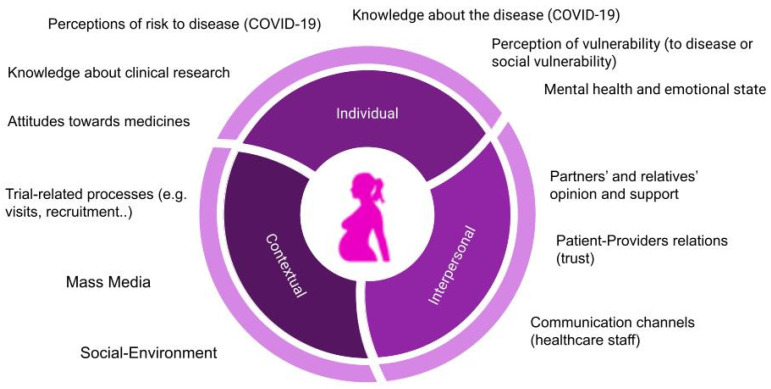
Pregnant women’s decision factors to participate in a clinical trial during pregnancy.

**Table 1 ijerph-18-10717-t001:** Sociodemographic characteristics of participants in the interviews.

	Pregnant Women (*N* = 24)		Healthcare Professionals (*N* = 6)	
	*n*	%	*n*	%
**Characteristics**				
**Female**	24	100	5	83.3
**Age (years)**				
<25	0	0	1	16.7
25–30	2	8.3	1	16.7
31–35	11	45.8	0	0
36–40	8	33.3	2	33.3
>40	3	12.5	3	50.0
**Nationality**				
Spanish	21	87.5	5	83.3
Brazilian	1	4.2	0	0
Bolivian	1	4.2	0	0
Peruvian	1	4.2	0	0
Mexican	0	0	1	16.7
**Education**				
Primary	1	4.2	0	0
Secondary	0	0	0	0
Practical training	2		0	0
University	19	79.2	6	100
**Occupation**				
Employed full-time	15	62.5	6	100
Unemployed	6	25.0	0	0
Self-employed (freelance)	1	4.2	0	0
Employed, currently on pregnancy leave	2	8.3	0	0
**Religion**				
Christian	2	12.5	1	16.7
None	21	87.5	5	83.3
**COVID-19 status**				
COVID-19 confirmed infection	7	29.2	1	16.7
Close contact with a COVID-19 case	7	29.2	5	83.3
**Healthcare providers’ characteristics**				
Medical doctor	NA		4	66.7
Nurse	NA		2	33.3
**Years working at health center**				
<5	NA		2	33.3
5 or more	NA		5	66.7

NA: Not applicable.

**Table 2 ijerph-18-10717-t002:** Acceptability of participating in a COVID-19 clinical trial reported among pregnant women and healthcare professionals.

Question	Pregnant Women (*N* = 24)		Healthcare Professionals (*N* = 6)	
	*n*	%	*n*	%
**Would you participate in a clinical trial as a non-pregnant adult?**				
Yes	7	29.2	4	66.7
No	9	37.5	1	16.7
Maybe	8	33.3	1	16.7
**Would you participate in a clinical trial during pregnancy?**				*
Yes	4	16.7	2	40.0
No	14	58.3	1	20.0
Maybe	6	25.0	2	40.0
**Would you participate in a clinical trial during breastfeeding?**		**		*
Yes	3	14.3	4	80.0
No	12	57.1	0	0
Maybe	6	28.6	1	20.0
**Would you participate in a clinical trial while pregnant in which HCQ is the study drug?**				*
Yes	5	20.8	4	80.0
No	15	62.5	1	20.0
Maybe	4	16.7	0	0
**Would you participate in a clinical trial while pregnant in which HCQ is NOT the study drug?**				*
Yes	5	20.8	2	40.0
No	15	62.5	1	20.0
Maybe	4	16.7	2	40.0

HCQ: hydroxychloroquine; * only applicable to female healthcare professionals (*N* = 5); ** in one interview, this topic was not raised, and two women were not able to lactate due to medical reasons (*N* = 21).

**Table 3 ijerph-18-10717-t003:** Reasons given by interviewed pregnant women for participating or not in a clinical trial.

Code	Reason
**Acceptable**	To help others She assumes that a clinical trial with pregnant women is reliable/safe If not pregnant, and the clinical trial is for her direct benefit for a disease she already has She already knows the drug because she is taking HCQ (during pregnancy) due to other medical concerns Empathy with healthcare staff who have been working during the pandemic To advance in science
**Not acceptable**	She does not like taking drugs on a general basis She does not like taking drugs during pregnancy She has been told to not take drugs during pregnancy She is not infected with the disease under study Uncertainties and fear about the consequences of the drug, or the placebo, on herself and her baby Because a drug that has to be tested and it may not be safe Perception of HCQ as a non-effective drug Not receiving clear information about the clinical trial by a medical doctor She is already taking other drugs during pregnancy (to avoid interactions) She forgets about taking pills regularly She just wants to be calmed and relaxed during pregnancy, does not want to have extra medical follow-ups Not acceptable by her partner/relatives Because there is not a higher risk of the disease for pregnant women Still too early to trust scientific articles and studies Own decision may affect her baby
**Circumstances for doubting or accepting to participate in a trial**	If it does not have adverse effects (does not harm the baby) During breastfeeding, because the baby is already born and healthy If this drug has already been tested during pregnancy If a medical doctor explains why it is positive for current pregnancy (clear benefit for current pregnancy) If there is evidence that COVID-19 causes a severe fetal disease If there’s a higher risk of contagion and dangerous consequences among pregnant women If relatives/partner agree (or not) with her participation in the clinical trial If there is a horrible situation for the COVID-19 pandemic (end-of-the-world-like) If she is infected and COVID-19 has severe risks for pregnancy

HCQ: hydroxychloroquine.

**Table 4 ijerph-18-10717-t004:** Barriers and facilitators for pregnant women to participate in a clinical trial during pregnancy.

Categories	Barriers	Facilitators
**Individual**	Attitudes of rejection towards drugs in general Attitudes of rejection towards any intervention, especially drugs or vaccines, during pregnancy Lack of knowledge of the disease under study Low perception of risk of disease severity	Willingness to help other pregnant women Feelings of contribution to advances in science High perception of risk and disease severity during pregnancy Awareness of the negative consequences of the disease on the fetus
**Interpersonal**	Absence of partner/relatives’ support for enrolling in the clinical trial	Partner and relatives’ presence and support for enrolling in the trial Close relationship with a healthcare professional who is trusted and recommends her inclusion
**Trial-related**	Not clear information about the intervention (drug/vaccine) of the clinical trial Possibility of avoiding infection by social isolation, without the need to take a pill Lack of trusted and/or known staff at recruitment Having to participate without being infected (for the purpose of prevention) Excess of the medical check-ups (trial appointments)	Adequate communication channels (intended time and setting) Information about the difference between the secondary/after effects and adverse effects of the study drug Understanding of the lack of adverse effects of the intervention in her fetus. Having to participate in being infected (for the purpose of treatment) Involvement of specialized clinical staff (gynecologists, etc.) at recruitment

## Data Availability

No new data were created or analyzed in this study. Data sharing is not applicable to this article.

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
