# Peer review of "Acceptability of Clinical Trials on COVID-19 during Pregnancy among Pregnant Women and Healthcare Providers: A Qualitative Study"

_ijerph, 2021, doi:10.3390/ijerph182010717_

Round 1
Reviewer 1 Report
The manuscript is a qualitative study about the acceptability and perceptions of clinical trials on COVID-19 during pregnancy among pregnant women and healthcare professionals.
The subject is ethically, scientifically and gendering sensitive and necessary. In spite of the authors highlighted the contribution of the study is "to help improving strategies for engagement of pregnant women in clinical trials"(page 2, line 85-6), I think the study will also contribute to analyse crucial issues involved as the imaginary of motherhood and pregnancy, ethical concerns, science and gender implications, among others.
The study was very well structured with a short bibliographic review about the theme, organized and adequate fieldwork among pregnant women and health workers.
The analysis of the interviews and discussion are very rich and enough based on the empirical data and narratives, although I think it could be enriched by more theoretical inputs.
The only point that I would like to emphasise is the choice of a study of HCQ could contribute to being a more sensitive arena than if the authors choose another drug with fewer public controversies.
In spite of women and health professionals didn't mention the industry interests, it would be adequate to mention in the introductory words the economy and for-profit bias in the clinical trials.
I strongly recommend the publication of this study and congratulate the authors for the careful work of written and analysis.
Author Response
REVIEWER 1
Authors’ responses:
Comment 1: The manuscript is a qualitative study about the acceptability and perceptions of clinical trials on COVID-19 during pregnancy among pregnant women and healthcare professionals.
The subject is ethically, scientifically and gendering sensitive and necessary. In spite of the authors highlighted the contribution of the study is "to help improving strategies for engagement of pregnant women in clinical trials"(page 2, line 85-6), I think the study will also contribute to analyse crucial issues involved as the imaginary of motherhood and pregnancy, ethical concerns, science and gender implications, among others.
Response: We would like to thank the reviewer for this very positive feedback. We agree with the reviewer that the study’s contribution may go beyond clinical-trials related issues. In response to this comments, we have enriched the manuscript with the excellent suggestion provided, as introduced now in the Discussion section (lines 454-457) “Finally, by providing relevant insights on how pregnant women relate to clinical research, the study also points at crucial issues to analyse conceptualizations of motherhood and pregnancy, or ethics in research during pregnancy“
Comment 2: The study was very well structured with a short bibliographic review about the theme, organized and adequate fieldwork among pregnant women and health workers.
Response: Thank you very much for your positive feedback and recognition of our work.
Comment 3: The analysis of the interviews and discussion are very rich and enough based on the empirical data and narratives, although I think it could be enriched by more theoretical
Response: We agree with the reviewer that the analysis could be improved by providing more theoretical inputs. Following his/her advice, we have expanded the Discussion section. The revised text reads as follows (lines 352-363): Our findings point at several factors that influence the decision-making process to participate in clinical trials during pregnancy, which, in turn, dialogue with the multidimensional approach to acceptability that has informed our study (28). The theoretical framework proposed by Sekhon et al. (28) envisages seven component constructs comprising a complex conceptualization of acceptability (either anticipated or reported). Even if in our study factors have been grouped in broader categories, certain parallels can be found between the identified potential barriers and facilitators, and the seven constructs that build up the definition of acceptability in the aforementioned theoretical framework. For instance, the affective attitude and the burden associated with participating in the intervention are reflected in our participants’ insights. Therefore, our results support the need to adopt a multidimensional perspective to provide a more nuanced assessment of the acceptability to participate in clinical trials involving pregnant women.
Comment 4: The only point that I would like to emphasise is the choice of a study of HCQ could contribute to being a more sensitive arena than if the authors choose another drug with fewer public controversies.
Response: Regarding the use of HCQ or any less controversial drug, this study was carried out as a qualitative study, ancillary to a clinical trial from the same research group, where HCQ was being tested for its safe use during pregnancy. This is why in this qualitative study, some questions were asked about HCQ or other drugs, but the name of the drug did not sound familiar to most participants, thus, it did not influence acceptability to participate in a clinical trial during pregnancy in the COVID-19 pandemic context.
Comment 5: In spite of women and health professionals didn't mention the industry interests, it would be adequate to mention in the introductory words the economy and for-profit bias in the clinical trials.
Response: We appreciate the recommendation of mentioning the impact of the pharmaceutical industry and the economy and for-profit bias in clinical trials, and we agree that it would have been interesting to raise this important topic in the background of our study. However, the authors think that it seems slightly out of the scope of the topic that should be addressed in the introduction. Our aim was to approach the acceptability of clinical trials among pregnant women by focusing on the decision-making processes, thus an overview centered on self-perceptions identified in existing studies was deemed more appropriate to contextualise the purpose of our study, rather than providing an explanation on the dynamics behind clinical research in general. Authors think that mentioning the impact of the pharmaceutical industry would require a more in-depth discussion later in the text, and would not be articulated with the factors identified as influencing the acceptability and the decision-making process to participate in clinical trials during pregnancy from a phenomenological perspective.
Comment 6: I strongly recommend the publication of this study and congratulate the authors for the careful work of written and analysis.
Response: Thank you, we really appreciate this encouraging comment.
We hope these improvements in the manuscript will meet reviewers’ expectations.
Kind regards,
Reviewer 2 Report
The authors presented a balanced document, scientifically sounded and relevant to current health conditions affecting the globe. I have several recommendations to increase the contribution.
- Check English proofreading. Example lines 58, 62, 280,
- Table 2 needs revisions. Some numbers lack period.
- Move line 252 earlier in the document.
- Consider revising figure 1 since it image does not express the dimensions of the identified barriers and facilitators or different factors that emerged as affecting the decision-making process of pregnant women to participate in a clinical trial. The figure does not help to discriminate if factors have the same impact or not if they weigh the same for the participant's decision-making or areas they rate on relevancy or importance one from another. Are they all at the same level?
Author Response
REVIEWER 2
Authors’ responses:
Comment 1: The authors presented a balanced document, scientifically sounded and relevant to current health conditions affecting the globe. I have several recommendations to increase the contribution.
Response: Thank you very much for your positive feedback and recognition of the relevance of our contribution. Following your advice, we are submitting a revised version of the manuscript. Herein, there is a detailed response to your comments and suggestions.
Comment 2: Check English proofreading. Example lines 58, 62, 280,
Response: In response to these comments, we have done English proofreading and several corrections have been made throughout the manuscript (including the ones highlighted by the reviewer).
Comment 3: Table 2 needs revisions. Some numbers lack period.
Response: Thanks for your observation. Table 2 has been revised and updated (Page 8).
Comment 4: Move line 252 earlier in the document.
Response: We appreciate this suggestion. The quote in line 252, has been moved to line 249, as suggested by the reviewer.
Comment 5: Consider revising figure 1 since its image does not express the dimensions of the identified barriers and facilitators or different factors that emerged as affecting the decision-making process of pregnant women to participate in a clinical trial. The figure does not help to discriminate if factors have the same impact or not if they weigh the same for the participant's decision-making or areas they rate on relevancy or importance one from another. Are they all at the same level?
Response: We apologize if our original Figure 1 did not properly show the dimensions that emerged as factors influencing the decision-making process of pregnant women to participate in a clinical trial. We certainly agree with the reviewer that the figure may not be as explicative as it should be, yet we think that all identified factors are interrelated, and their impact may depend on this interrelation. Therefore, we have preferred not to visually differentiate their relevance. We have modified the figure (page 12), in order to better capture the barriers and facilitators identified and hope that now it helps provide a clear overview of the factors influencing decision-making processes.
We hope that these improvements in the manuscript will meet reviewers’ expectations.
Kind regards,
Reviewer 3 Report
Elena et. al. studied the acceptability to participate in COVID-19 clinical trials among pregnant women. In the present manuscript, the authors interviewed pregnant women and healthcare providers, and built the linkage among them. The information is interesting and useful for developing relation between COVID-19 and health impact on pregnant women. It’s well designed and written. For these reasons, this manuscript is suitable for publishing in International Journal of Environmental Research and Public Health, but needs minor subsequent revision. Some comments/suggestions are listed.
- The format of ref needs to be revised.
- Please pay attention to the significant digits of the result listed in Tables.
Author Response
REVIEWER 3:
Authors’ responses:
Comment 1: Elena et. al. studied the acceptability to participate in COVID-19 clinical trials among pregnant women. In the present manuscript, the authors interviewed pregnant women and healthcare providers, and built the linkage among them. The information is interesting and useful for developing relation between COVID-19 and health impact on pregnant women. It’s well designed and written. For these reasons, this manuscript is suitable for publishing in International Journal of Environmental Research and Public Health, but needs minor subsequent revision. Some comments/suggestions are listed.
Response: We would like to thank the reviewer for this encouraging comment and recognition of the relevance of our contribution. In light of the reviewer’s all suggestions, the manuscript has been revised.
Comment 2: The format of ref needs to be revised.
Response: Following reviewer’s suggestion, we have revised the format of the references.
Comment 3: Please pay attention to the significant digits of the result listed in Tables.
Response: Thank you for pointing this out. The reviewer is correct, tables have been revised and corrected accordingly.
We hope that these improvements in the manuscript will meet reviewers’ expectations.
Kind regards,
Reviewer 4 Report
- Although the manuscript is interesting, there are some details that need to be clarified:
1. What really is the rationale for this research? what knowledge gap does it fill? Why is this a public health issue? Why should it be published in a public health magazine? this is not clear
2. The method is not accurate:
The analysis process and results should be described in minimum and sufficient detail so that readers have a clear understanding of how the analysis was carried out, its strengths and limitations. -
How were participants selected? e.g. purposive, convenience, consecutive, snowball
How many people refused to participate or dropped out? Reasons?
Was anyone else present besides the participants and researchers?
Were questions, prompts, guides provided by the authors? Was it pilot tested?
Were repeat interviews carried out? If yes, how many?
What was the duration of the interviews?
Was data saturation discussed?
Were themes identified in advance or derived from the data?
Did participants provide feedback on the findings?
The description of the data collection and analysis process contemplates a coming and going that confuses the reader. Despite describing the text, it lacks basic information that gives meaning to the results presented. It is general information that is not linked to the studied content.
Finally, why were the results presented this way? How were these speeches selected?
Author Response
REVIEWER 4:
Authors’ responses
Comment 1: Although the manuscript is interesting, there are some details that need to be clarified.
Response: We would like to thank the reviewer for his/her feedback. Following the reviewer's comments, we are submitting a revised version of the manuscript. Herein, there is a detailed response to his/her comments and suggestions.
Comment 2: What really is the rationale for this research? what knowledge gap does it fill? Why is this a public health issue? Why should it be published in a public health magazine? this is not clear
Response: We appreciate this comment. While we agree that it is essential to demonstrate that the study fills a knowledge gap within the Public Health domain, we firmly think that our research makes a valuable contribution to the field. In the Introduction section, we have made an effort to present the limited inclusion of pregnant women in clinical trials of medicinal products (lines 48-62). In lines 58-59 we have specifically mentioned recent initiatives advocating for the inclusion of pregnant women in drug and vaccine trials during the COVID-19 pandemic. Given that clinical trials during pregnancy are scarce, issues related to the acceptability to participate, factors influencing decision-making processes, or how pregnant women relate to clinical research are, therefore, underexplored. To further contextualise the limited evidence on this topic, in lines 79-83 we mention the need to understand pregnant women’s perceptions and experiences with regard to clinical trials in order to develop evidence-based recommendations to improve engagement of pregnant women in clinical research.
In the introduction, we also made reference to some works focused on this topic. However, the COVID-19 pandemic provides a novel context. On the one hand, by reporting insights and first-hand experiences from women who were pregnant in the first and/or second waves of the COVID-19 pandemic in one of the countries with the highest prevalence of infection (Spain), we think that the study necessarily contributes with novel and original findings. On the other hand, in lines 79-80 we highlighted the lack of clinical trials on investigational medicinal products for COVID-19 treatment and prevention during pregnancy. Therefore, the evidence brought by the study contributes to filling an important gap regarding pregnant women’s perceptions about participation in clinical research, especially, but not restricted to drug-based clinical trials within the frame of a public health emergency such as the COVID-19 pandemic. This evidence should provide useful information to improve recruitment strategies for engaging pregnant women in clinical trials (line 92). Finally, the study may also contribute to expanding the knowledge about perceptions around motherhood and pregnancy and related ethical concerns with regard to clinical science and, indeed, from a social science perspective.
Having said that, we agree with the reviewer that we could have stressed the rationale of the study and its contribution. The study aim has been modified to expand the contribution of the paper to the identified research gap. Please see lines (lines 92-93): The study may contribute to the understanding of the impact of COVID-19 on women pregnant during the pandemic. An additional comment has been introduced in lines 454-457: “Finally, by providing relevant insights on how pregnant women relate to clinical research, the study also points at crucial issues to analyse conceptualizations of motherhood and pregnancy, or ethics in research during pregnancy”.
Comment 3. The method is not accurate:
The analysis process and results should be described in minimum and sufficient detail so that readers have a clear understanding of how the analysis was carried out, its strengths and limitations.
Response: We acknowledge the relevance of the aspect raised by the reviewer. In the methods section, the analysis process was already detailed (please see line 138). The strengths and limitations of the study were already mentioned by the end of the manuscript (please see line 444).
Revising the issues he/she has asked to clarify below, we assume that he/she is referring to some of the items stated in the COREQ guidelines for reporting qualitative research (https://academic.oup.com/intqhc/article/19/6/349/1791966), even though there’s not an explicit reference to it in his/her review. In this respect, authors would like to mention that another applicable reporting guideline has been chosen to prepare the article, namely the SRQR guidelines (https://www.equator-network.org/reporting-guidelines/srqr/), and the manuscript contains all relevant items accordingly. For the sake of transparency, a completed reporting checklist from SRQR guidelines has been now submitted as an additional file. In the methods section from the revised manuscript, a specific reference to it has been introduced (lines 146-147): This article has been prepared as per reporting standards set forth in the SRQR guidelines for reporting qualitative studies.
Nevertheless, we are unsure as to whether the reviewer is suggesting or claiming that a completed reporting checklist from COREQ guidelines should be included as an additional file in our revised manuscript, since there isn’t a specific demand in this regard. However, while we hope that the completed reporting checklist using the SRQR guidelines included now as an additional file may meet his/her expectations, we will address his/her concerns as follows:
- How were participants selected? e.g. purposive, convenience, consecutive, snowball. In the original manuscript we had already described the different sampling strategies used in our study, as stated in lines 108-111 (currently 116-119), “Initial sampling was purposive, and snowball sampling was also applied to interview other pregnant women, not necessarily attending study hospitals. In addition, convenience sampling was applied to facilitate access to women willing to be interviewed”.
- How many people refused to participate or dropped out? Reasons? No participant refused to be interviewed, and no participant dropped out.
- Was anyone else present besides the participants and researchers? During the interviews conducted along the study, the only people present were the interviewer and the interviewee. In the original manuscript, we stated that: Two researchers performed the interviews (E.M-C. and C.E-F.). Each interview was performed by one of them” (lines 130-132). Additionally, it must be recalled that interviews were performed online and/or by telephone due to restrictive and preventive measures against COVID-19.
- Were questions, prompts, guides provided by the authors? Was it pilot tested? A semi-structured interview question guide was not provided. This has been now submitted as an additional file in the reviewed manuscript. This interview guide was pilot-tested once to check its appropriateness, interviewing style and approach. We have now specified it in the manuscript and now it reads: The topic guide was pilot-tested to check the interviewing style and approach (lines 127-128).
- Were repeat interviews carried out? If yes, how many? Given that the study was not conceived to be longitudinal, we confirm that repeated interviews were not planned nor actually conducted.
- What was the duration of the interviews? The duration of the interviews was specified in the original manuscript. Actually in lines 119-120 (current lines 127-128 in the reviewed manuscript), it is stated that “Interviews were performed in Spanish or Catalan, and lasted approximately 20 to 40 minutes each”.
- Was data saturation discussed? Concerning the data saturation discussion, in the original manuscript, we mentioned that “The sample size was defined based on saturation point, whereby all themes had been thoroughly explored and no new themes kept emerging in subsequent interviews” (lines 117-119, now 119-121) and “Then, consensus on codes and emerging themes were reached between two investigators (E.M-C. and C.E-F.) A final coding frame was agreed, and themes and categories were grouped with verbatim quotes” (lines 140-143, current 142-145 in the reviewed manuscript).
- Were themes identified in advance or derived from the data? We believe that this concern was addressed in the above comment. Additionally, we would like to clarify that, according to the inductive perspective provided by Grounded theory (one of the methodological approaches used in the study -see line 98, within the “Study design” subsection), no hypotheses or theoretical frameworks were previously adopted. Therefore, neither themes were identified in advance nor any conceptual framework was used during the analysis process.
- Did participants provide feedback on the findings? We think that this is an important point. However, a participatory approach, that would actually envisage the discussion of results with participants, was not applied as part of the methodological approaches.
Comment 4: The description of the data collection and analysis process contemplates a coming and going that confuses the reader. Despite describing the text, it lacks basic information that gives meaning to the results presented. It is general information that is not linked to the studied content.
Response: We are not fully clear regarding what the reviewer means with this comment. We did not make adjustments in the hope that the reviewer clarifies his/her point, so that we can address it accordingly. Having stated that, we have made an effort to answer his/her concerns:
- Either data collection and data analysis processes are independently described and, indeed, presented in separate subsections.
- We are unsure as to what kind of basic information is lacking that would help to give meaning to the results. We hope that by adding the interview guide as a supplementary file we managed to address this concern.
Comment 5: Finally, why were the results presented this way? How were these speeches selected?
Response: The results are presented according to usual standards for reporting qualitative research (see our previous response with regard to SRQR guidelines), where it is recommended to provide links with empirical data (in this case quotes from interviews transcribed verbatim) that support the analytic findings. After in-depth analysis, codes emerged, and quotes were selected based on their representativeness within the codes, or their richness of the speech. This is not only a common practice, but a standardized method to present qualitative results.
We hope to have addressed reviewers' concerns and meet his/her expectations.
Kind regards,
Round 2
Reviewer 4 Report
The authors complied with all recommendations and the manuscript is suitable for publication.